# Data Cleansing for Models Trained with SGD

**Satoshi Hara**[*]       **Atsushi Nitanda**[†]       **Takanori Maehara**[‡]

## Abstract

Data cleansing is a typical approach used to improve the accuracy of machine learning models, which, however, requires extensive domain knowledge to identify the influential instances that affect the models. In this paper, we propose an algorithm that can identify influential instances without using any domain knowledge. The proposed algorithm automatically cleans the data, which does not require any of the users' knowledge. Hence, even non-experts can improve the models. The existing methods require the loss function to be convex and an optimal model to be obtained, which is not always the case in modern machine learning. To overcome these limitations, we propose a novel approach specifically designed for the models trained with stochastic gradient descent (SGD). The proposed method infers the influential instances by retracing the steps of the SGD while incorporating intermediate models computed in each step. Through experiments, we demonstrate that the proposed method can accurately infer the influential instances. Moreover, we used MNIST and CIFAR10 to show that the models can be effectively improved by removing the influential instances suggested by the proposed method.

## 1 Introduction

Building accurate models is one of the fundamental goals in machine learning. If the obtained model is not satisfactory, users try to improve the model in several ways such as by modifying input features, cleansing data, or even by gathering additional data. Error analysis [Ng, 2017] is a typical approach for this purpose. In this analysis, the users hypothesize the cause of model's failure by investigating important features or examining the misclassified instances. However, a good hypothesis requires experience and domain knowledge. Therefore, it is difficult for non-domain experts or non-machine learning specialists to build accurate models.

How can we help non-experts to build accurate machine learning models? In this study, we focus on the following data cleansing problem that removes "harmful" instances from the training set.

**Problem 1** (Data Cleansing). Find a subset of the training instances such that the trained model obtained after removing the subset has a better accuracy.

Currently, the users hypothesize the training instances that can have certain influences on the resulting models by inspecting instances based on the domain knowledge. Our aim is to develop an algorithm that can identify influential instances *without using any domain knowledge*. With such an algorithm, the users do not need to hypothesize influential instances. Instead, the algorithm automatically cleans the data, which does not require any of the users' knowledge. Hence, with this process, even non-experts can improve the models.

For data cleansing, we need to determine the training instances that affect the model. In the literature of statistics, an *influential instance* is defined as the instance that leads to a distinct model from the current model if the corresponding instance is absent [Cook, 1977]. A naive approach to determine

---

[*]`satohara@ar.sanken.osaka-u.ac.jp`, Osaka University, Japan

[†]`nitanda@mist.i.u-tokyo.ac.jp`, The University of Tokyo, Japan

[‡]`takanori.maehara@riken.jp`, RIKEN AIP, Japan

these influential instances is, therefore, to retrain the model by leaving every one instance out of the training set, which can be computationally very demanding. To efficiently infer an influential instance without retraining, the convexity of the loss function plays an important role. Pioneering studies by Beckman and Trussell [1974], Cook [1977], and Pregibon [1981] have shown that, for some convex loss functions, the influential instances can be inferred without model retraining by utilizing the optimality condition on the training loss, given that an optimal model is obtained. A recent study by Koh and Liang [2017] further generalized these approaches to any smooth and strongly convex loss functions by incorporating the idea of influence function [Cook and Weisberg, 1980] in robust statistics (see Section 6).

The focus of this study is to go beyond the convexity and optimality. We aim to develop an algorithm that can infer influential instances even for non-convex objectives such as deep neural networks. To this end, we propose a completely different approach to infer the influential instances. The proposed approach is based on the stochastic gradient descent (SGD). Modern machine learning models including deep neural networks are trained using SGD and its variants. Our idea is to redefine the notion of influence for the models trained with SGD, which we named *SGD-influence*. Based on SGD-influence, we propose a method that infers the influential instances without model retraining. The proposed method is based solely on the analysis of SGD. Different from the existing methods, the proposed method does not require the optimality conditions to hold true on the obtained models. The proposed method is therefore suitable to the SGD context where we no longer look for the exact optimum of the training loss. In SGD, we instead look for the minimum error on the validation set, which leads to early stopping of the optimization that can violate the optimality condition.

In summary, the contribution of this study is threefold.

- We propose a new definition of the influence, which we name as *SGD-influence*, for the models trained with SGD. SGD-influence is defined based on the counterfactual effect: what if an instance is absent in SGD, how largely will the resulting model change?
- We propose a novel estimator of SGD-influence based on the analysis of SGD. We then construct a proposed influence estimation algorithm based on this estimator. We also study the estimation error of the proposed estimator on both convex and non-convex loss functions.
- Through experiments, we demonstrate that the proposed method can accurately infer the influential instances. Moreover, we used MNIST and CIFAR10 to show that the models can be effectively improved by removing the influential instances suggested by the proposed method.

## 2 Preliminaries

**Notations** For vectors $a, b \in \mathbb{R}^p$, we denote the inner product by $\langle a, b \rangle = \sum_{i=1}^p a_i b_i$, and the norm by $\|a\| = \sqrt{\langle a, a \rangle}$. For a function $f(\theta)$ with $\theta \in \mathbb{R}^p$, we denote its derivative by $\nabla_\theta f(\theta)$.

**Supervised Learning** Let $z = (x, y) \in \mathbb{R}^d \times \mathcal{Y}$ be an observation, which is a pair of $d$-dimensional input feature vector $x$ and output $y$ in a certain domain $\mathcal{Y}$ (e.g., $\mathcal{Y} = \mathbb{R}$ for regression, and $\mathcal{Y} = \{-1, 1\}$ for binary classification). The objective of learning is to find a model $f(x; \theta)$ that well approximates the output as $y \approx f(x; \theta)$. Here, $\theta \in \mathbb{R}^p$ is a parameter of the model.

Let $D := \{z_n = (x_n, y_n)\}_{n=1}^N$ be a training set with independent and identically distributed observations. We denote the loss function for an instance $z$ with the parameter $\theta$ by $\ell(z; \theta)$. The learning problem is then denoted as

$$\hat{\theta} = \operatorname{argmin}_{\theta \in \mathbb{R}^p} \frac{1}{N} \sum_{n=1}^N \ell(z_n; \theta). \tag{1}$$

**SGD** Let $g(z; \theta) := \nabla_\theta \ell(z; \theta)$. SGD starts the optimization from the initial parameter $\theta^{[1]}$. An update rule of the mini-batch SGD at the $t$-th step for the problem (1) is given by $\theta^{[t+1]} \leftarrow \theta^{[t]} - \frac{\eta_t}{|S_t|} \sum_{i \in S_t} g(z_i; \theta^{[t]})$, where $S_t$ denotes the set of instance indices used in the $t$-th step, and $\eta_t > 0$ is the learning rate. We denote the number of total SGD steps by $T$.

## 3 SGD-Influence

We propose a novel notion of influence for the models trained with SGD, which we name as *SGD-influence*. We then formalize the influence estimation problem we consider in this paper.

We define SGD-influence based on the following *counterfactual* SGD where one instance is absent.

**Definition 2** (Counterfactual SGD). The counterfactual SGD starts the optimization from the same initial parameter as the ordinary SGD $\theta_{-j}^{[1]} = \theta^{[1]}$. The $t$-th step of the counterfactual SGD with the $j$-th instance $z_j$ absent is defined by $\theta_{-j}^{[t+1]} \leftarrow \theta_{-j}^{[t]} - \frac{\eta_t}{|S_t|} \sum_{i \in S_t \setminus \{j\}} g(z_i; \theta_{-j}^{[t]})$.

**Definition 3** (SGD-Influence). We refer to the parameter difference $\theta_{-j}^{[t]} - \theta^{[t]}$ as the *SGD-influence* of the instance $z_j \in D$ at step $t$.

It should be noted that SGD-influence can be defined in every step of SGD, even for non-optimal models. Thus, SGD-influence is a suitable notion of influence for the cases where we no longer look for the exact optimal of (1). In this study, we specifically focus on estimating an inner product of a query vector $u \in \mathbb{R}^p$ and the SGD-influence after $T$ SGD steps, as follows.

**Problem 4** (Linear Influence Estimation (LIE)). For a given query vector $u \in \mathbb{R}^p$, estimate the *linear influence* $L_{-j}^{[T]}(u) := \langle u, \theta_{-j}^{[T]} - \theta^{[T]} \rangle$.

LIE includes several important applications (see [Koh and Liang, 2017]). One important application is the influence estimation on the loss. If we take $u = \nabla_\theta \ell(x; \theta^{[T]})$ for an input $x$, LIE amounts to estimating the change in loss $L_{-j}^{[T]}(\nabla_\theta \ell(x; \theta^{[T]})) \approx \ell(x; \theta_{-j}^{[T]}) - \ell(x; \theta^{[T]})$. Negative $L_{-j}^{[T]}(\nabla_\theta \ell(x; \theta^{[T]}))$ indicates that the loss on the input $x$ can be decreased by removing $z_j$.

Note that SGD-influence as well as linear influence can be computed exactly by running the counterfactual SGD for all $z_j \in D$. However, this requires running SGD $N$ times, which is computationally demanding even for $N \approx 100$. Therefore, our goal is to develop an estimation algorithm for LIE, which does not require running SGD multiple times.

# 4 Estimating SGD-Influence

In this section, we present our proposed estimator of SGD-influence and show its theoretical properties. We then derive an algorithm for LIE based on the estimator in the next section.

## 4.1 Proposed Estimator

We estimate SGD-influence using the first-order Taylor approximation of the gradient. Here, we assume that the loss function $\ell(z; \theta)$ is twice differentiable. We then obtain $\frac{1}{|S_t|} \sum_{i \in S_t} \left( \nabla_\theta \ell(z_i; \theta_{-j}^{[t]}) - \nabla_\theta \ell(z_i; \theta^{[t]}) \right) \approx H^{[t]}(\theta_{-j}^{[t]} - \theta^{[t]})$, where $H^{[t]} := \frac{1}{|S_t|} \sum_{i \in S_t} \nabla_\theta^2 \ell(z_i; \theta^{[t]})$ is the Hessian of the loss on the mini-batch $S_t$. With this approximation, denoting an identity matrix by $I$, we have

$$\theta_{-j}^{[t]} - \theta^{[t]} = (\theta_{-j}^{[t-1]} - \theta^{[t-1]}) - \frac{\eta_{t-1}}{|S_{t-1}|} \sum_{i \in S_{t-1}} (\nabla_\theta \ell(z_i; \theta_{-j}^{[t-1]}) - \nabla_\theta \ell(z_i; \theta^{[t-1]}))$$

$$\approx (I - \eta_{t-1} H^{[t-1]})(\theta_{-j}^{[t-1]} - \theta^{[t-1]}).$$

We construct an estimator for the SGD-influence based on this approximation. For simplicity, here, we focus on one-epoch SGD where each instance appears only once. Let $Z_t := I - \eta_t H^{[t]}$ and $\pi(j)$ be the SGD step where the instance $z_j$ is used. By recursively applying the approximation and recalling that $\theta_{-j}^{[\pi(j)+1]} - \theta^{[\pi(j)+1]} = \frac{\eta_{\pi(j)}}{|S_{\pi(j)}|} g(z_j; \theta^{[\pi(j)]})$, we obtain the following estimator

$$\theta_{-j}^{[T]} - \theta^{[T]} \approx \frac{\eta_{\pi(j)}}{|S_{\pi(j)}|} Z_{T-1} Z_{T-2} \cdots Z_{\pi(j)+1} g(z_j; \theta^{[\pi(j)]}) =: \Delta\theta_{-j}. \tag{2}$$

## 4.2 Properties of $\Delta\theta_{-j}$

Here, we evaluate the estimation error of the proposed estimator $\Delta\theta_{-j}$ for both convex and non-convex loss functions. A notable property of the estimator $\Delta\theta_{-j}$ is that, unlike existing methods, the error can be evaluated *even without assuming the convexity of the loss function* $\ell(z; \theta)$.

**Convex Loss** For smooth and strongly convex problems, there exists a uniform bound on the gap between the SGD-influence $\theta_{-j}^{[T]} - \theta^{[T]}$ and the proposed estimator $\Delta\theta_{-j}$.

**Theorem 5.** Assume that $\ell(z;\theta)$ is twice differentiable with respect to the parameter $\theta$ and there exist $\lambda, \Lambda > 0$ such that $\lambda I \prec \nabla_\theta^2 \ell(z;\theta) \prec \Lambda I$ for all $z, \theta$. If $\eta_s \leq 1/\Lambda$, then we get

$$\|(\theta_{-j}^{[T]} - \theta^{[T]}) - \Delta\theta_{-j}\| \leq \sqrt{2(h_j(\lambda)^2 + h_j(\Lambda)^2)}, \tag{3}$$

where $h_j(a) := \frac{\eta_{\pi(j)}}{|S_{\pi(j)}|}\prod_{s=\pi(j)+1}^{T-1}(1-\eta_s a)\|g(z_j;\theta^{[\pi(j)]})\|$.

**Non-Convex Loss** For non-convex loss functions, the aforementioned uniform bound no longer holds. However, we can still evaluate the growth of the estimation error. For simplicity, we consider a constant learning rate $\eta = O(\gamma/\sqrt{T})$ that depends only on the number of total SGD steps $T$. It should be noted that SGD with this learning rate is theoretically justified to converge to a stationary point [Ghadimi and Lan, 2013]. The next theorem indicates that $\Delta\theta_{-j}$ can approximate SGD-influence well if Hessian $\nabla_\theta^2 \ell(\theta, z)$ is Lipschitz continuous.

**Theorem 6.** Assume that $\ell(z;\theta)$ is twice differentiable and $\nabla_\theta^2 \ell(z;\theta)$ is $L$-Lipschitz continuous with respect to $\theta$. Moreover, assume that $\|\nabla_\theta \ell(z;\theta)\| \leq G$, $\nabla_\theta^2 \ell(z;\theta) \prec \Lambda I$ for all $z, \theta$. Consider SGD with a learning rate $\eta = O(\gamma/\sqrt{T})$. Then,

$$\|(\theta_{-j}^{[T]} - \theta^{[T]}) - \Delta\theta_{-j}\| \leq \frac{\exp^{O(\gamma\Lambda\sqrt{T})}\gamma^2 T G^2 L}{\Lambda}. \tag{4}$$

# 5 Proposed Method for LIE

We now derive our proposed method for LIE. First, we extend the estimator $\Delta\theta_{-j}$ to multi-epoch SGD. Let $\pi_1(j), \pi_2(j), \ldots, \pi_K(j)$ be the steps where the instance $z_j$ is used in $K$-epoch SGD. We estimate the effect of the step $\pi_k(j)$ based on (2) as $Z_{T-1}Z_{T-2}\cdots Z_{\pi_k(j)+1}\frac{\eta_{\pi_k(j)}}{|S_{\pi_k(j)}|}g(z_j;\theta^{[\pi_k(j)]})$. We then add all the effects and derive the estimator $\Delta\theta_{-j} = \sum_{k=1}^{K}\left(\prod_{s=1}^{T-\pi_k(j)-1}Z_{T-s}\right)\frac{\eta_{\pi_k(j)}}{|S_{\pi_k(j)}|}g(z_j;\theta^{[\pi_k(j)]})$.

Let $u^{[t]} := Z_{t+1}Z_{t+2}\ldots Z_{T-1}u$. LIE based on the estimator $\Delta\theta_{-j}$ is then obtained as

$$\langle u, \Delta\theta_{-j}\rangle = \sum_{k=1}^{K}\langle u^{[\pi_k(j)]}, \frac{\eta_{\pi_k(j)}}{|S_{\pi_k(j)}|}g(z_j;\theta^{[\pi_k(j)]})\rangle.$$

It should be noted that $u^{[t]}$ can be computed recursively $u^{[t]} \leftarrow Z_{t+1}u^{[t+1]} = u^{[t+1]} - \eta_{t+1}H_{\theta^{[t+1]}}u^{[t+1]}$ by retracing SGD. The proposed method is based on this recursive computation.

The proposed method consists of two phases, the training phase and the inference phase, as shown in Algorithms 1 and 2. In the training phase in Algorithm 1, during running SGD, we store the tuple of the instance indices $S_t$, learning rate $\eta_t$, and parameter $\theta^{[t]}$.[4] In the inference phase in Algorithm 2, we retrace the stored information and compute $u^{[t]}$ in each step.

---

**Algorithm 1** LIE for SGD: Training Phase

Initialize the parameter $\theta^{[1]}$
Initialize the sequence as null: $A \leftarrow \emptyset$
**for** $t = 1, 2, \ldots, T-1$ **do**
    $A[t] \leftarrow (S_t, \eta_t, \theta^{[t]})$   *// store information*
    $\theta^{[t+1]} \leftarrow \theta^{[t]} - \frac{\eta_t}{|S_t|}\sum_{i\in S_t}g(z_i;\theta^{[t]})$
**end for**

---

**Algorithm 2** LIE for SGD: Inference Phase

**Require:** $u \in \mathbb{R}^p$
Initialize the influence: $\hat{L}_{-j}^{[T]}(u) \leftarrow 0, \forall j$
**for** $t = T-1, T-2, \ldots, 1$ **do**
    $(S_t, \eta_t, \theta^{[t]}) \leftarrow A[t]$   *// load information*
    *// update the linear influence of $z_j$*
    $\hat{L}_{-j}^{[T]}(u) \mathrel{+}= \langle u, \frac{\eta_t}{|S_t|}g(z_j;\theta^{[t]})\rangle, \forall j \in S_t$
    $u \mathrel{-}= \eta_t H^{[t]}u$   *// update $u$*
**end for**

---

Note that, in Algorithm 2, we need to compute $H^{[t]}u^{[t]}$. A naive implementation requires $O(p^2)$ memory to store the matrix $H^{[t]}$, which can be prohibitive for very large models. We can avoid this difficulty by directly computing $H^{[t]}u^{[t]}$ without the explicit computation of $H^{[t]}$. Because $H^{[t]}u^{[t]} = \frac{1}{|S_t|}\sum_{i\in S_t}\nabla_\theta\langle u^{[t]}, \nabla_\theta\ell(z_i;\theta^{[t]})\rangle$, we only need to compute the derivative of $\langle u^{[t]}, \nabla_\theta\ell(z_i;\theta^{[t]})\rangle$, which does not require the explicit computation of $H^{[t]}$. For example, in Tensorflow, this can be implemented in a few lines.[5] The time complexity for the inference phase is $O(TM\delta)$, where $M$ is the largest batch size in SGD and $\delta$ is the complexity for computing the parameter gradient.

[tf.gradients(tf.tensordot(u, g, axes), theta) for g in grads], axis)```

## 6 Related Studies

**Influence Estimation** Traditional studies on influence estimation considered the change in the solution $\hat{\theta}$ to the problem (1) if an instance $z_j$ was absent. For this purpose, they considered the counterfactual problem $\hat{\theta}_{-j} = \operatorname{argmin}_\theta \sum_{n=1;n\neq j}^N \ell(z;\theta)$. The goal of the traditional influence estimation is to obtain an estimate of the difference $\hat{\theta}_{-j} - \hat{\theta}$ without retraining the models. Pioneering studies by Beckman and Trussell [1974], Cook [1977], and Pregibon [1981] have shown that the influence $\hat{\theta}_{-j} - \hat{\theta}$ can be computed analytically for linear and generalized linear models. Koh and Liang [2017] considered a further generalizations of those previous studies. They introduced the following approximation for strongly convex loss functions $\ell(z;\theta)$:

$$\hat{\theta}_{-j} - \hat{\theta} \approx \tfrac{1}{N}\hat{H}^{-1}\nabla_\theta \ell(z_j;\hat{\theta}), \tag{5}$$

where $\hat{H} = \frac{1}{N}\sum_{z\in D}\nabla^2\ell(z;\hat{\theta})$ is the Hessian of the loss for the optimal model. We note that Zhang et al. [2018] and Khanna et al. [2019] further extended this approach. Zhang et al. [2018] used this approach to fix the labels of the training instances. Khanna et al. [2019] proposed to find the influential instances using the Bayesian quadrature, which includes (5) as its special case.

Our study differs from these traditional approaches in two ways. First, the proposed SGD-influence does not assume the optimality of the obtained models. We instead consider the models obtained in each step of SGD, which are not necessarily optimal. Second, the proposed method does not require the function loss $\ell(z;\theta)$ to be convex. The proposed method is valid even for non-convex losses.

**Estimation of Data Importance** Some recent works [Ren et al., 2018; Ghorbani and Zou, 2019] focused on estimating the importance of each training instance. Ren et al. [2018] proposed weighting each training instance so that the validation loss to be minimized. Ghorbani and Zou [2019] introduced some axioms that the data importance should satisfy, and derived Shapley value as an ideal importance. These studies demonstrated the effectiveness of the proposed importances only empirically. The advantage of our study from these prior studies is in theories of the estimation error, that clarified in which circumstances the estimated importances are accurate.

**Learning from Noisy Labels** There are plenty of studies for training models from noisy labels [Aslam and Decatur, 1996; Brodley and Friedl, 1999; Natarajan et al., 2013; Zhang et al., 2018]. The difference from our study is that these studies assumed that the label noise is an only issue. However, as Figures 13 and 14 show, the model performance depends not only on label noises but atypical inputs also. For example, in Figure 13, we can find several atypical instances that even human cannot label them confidently. These atypical instances should be removed from the training rather than fixing the labels because we cannot put correct labels to them.

**Outlier Detection** A typical approach for data cleansing is outlier detection. Outlier detection is used to remove abnormal instances from the training set before training the model to ensure that the model is not affected by the abnormal instances. For tabular data, there are several popular methods such as One-class SVM [Schölkopf et al., 2001], Local Outlier Factor [Breunig et al., 2000], and Isolation Forest [Liu et al., 2008]. For complex data such as images, autoencoders can also be used [Aggarwal, 2016; Zhou and Paffenroth, 2017] along with generative adversarial networks [Schlegl et al., 2017]. It should be noted that although these methods can find abnormal instances, they are not necessarily influential to the resulting models, as we will show in the experiments.

## 7 Experiments

Here, we evaluate the two aspects of the proposed method: the performances of LIE and data cleansing. We used Python 3 and PyTorch 1.0 for the experiments.[6] The experiments were conducted on 64bit Ubuntu 16.04 with six Intel Xeon E5-1650 3.6GHz CPU, 128GB RAM, and four GeForce GTX 1080ti.

## 7.1 Evaluation of LIE

We first evaluate the effectiveness of the proposed method in the estimation of linear influence. For this purpose, we artificially created small datasets to ensure that the true linear influence is computable. The detailed setup can be found in Appendix C.1.

**Setup** We used three datasets: Adult [Dua and Karra Taniskidou, 2017], 20Newsgroups[7], and MNIST [LeCun *et al.*, 1998]. These are common benchmarks in tabular data analysis, natural language processing, and image recognition, respectively. We adopted these three datasets to demonstrate the validity of the proposed method across different data domains. For 20Newsgroups and MNIST, we selected the two document categories `ibm.pc.hardware` and `mac.hardware` and images from one and seven, respectively, so that the problem to be binary classification.

To observe the validity of the proposed method beyond convexity, we adopted two models, linear logistic regression and deep neural networks. For deep neural networks, we used a network with two fully connected layers with eight units each and ReLU activation. We used the sigmoid function at the output layer and adopted the cross entropy as the loss function. It should be noted that the loss function for linear logistic regression is convex, while that for deep neural networks is non-convex.

In the experiments, we randomly subsampled 200 instances for the training set $D$ and validation set $D'$. We then estimated the linear influence for the validation loss using Algorithm 2. Here, we set the query vector $u$ as $u = \frac{1}{|D'|} \sum_{z' \in D'} \nabla_\theta \ell(z'; \theta^{[T]})$. The estimation of linear influence thus amounts to estimating the change in the validation loss $\langle u, \theta^{[T]}_{-j} - \theta^{[T]} \rangle \approx \frac{1}{|D'|} \sum_{z' \in D'} \left( \ell(z'; \theta^{[T]}_{-j}) - \ell(z'; \theta^{[T]}) \right)$.

**Evaluation** We ran the counterfactual SGD for all $z_j \in D$ and computed the true linear influence. For evaluation, we compared the estimated influences with this true influence using Kendall's tau and Jaccard index. With Kendall's tau, a typical metric for ordinal associations, we measured the correlation between the estimated and true influences. Kendall's tau takes the value between plus and minus one, where one indicates that the orders of the estimated and true influences are identical. With Jaccard index, we measured the identification accuracy of the influential instances. For data cleansing, the users are interested in instances with large positive or negative influences. We selected ten instances with the largest positive and negative true influences and constructed a set of 20 important instances. We compared this important instances with the estimated ones using Jaccard index, which varies between zero and one, where the value one indicates that the sets are identical.

**Results** We adopted the method proposed by Koh and Liang [2017] in (5) as the baseline, abbreviated as K&L. For deep neural networks, the Hessian matrix is not positive definite, which makes the estimator (5) invalid. To alleviate the effect of negative eigenvalues, we added a positive constant 1.0 to the diagonal as suggested by Koh and Liang [2017].

Figure 1 shows a clear advantage of the proposed method. The proposed method successfully estimated the true linear influences with high precision. The estimated influences were concentrated on the diagonal lines, indicating that the estimated influences accurately approximated the true influences. In contrast, the estimated influences obtained by K&L were less accurate. We observed that the estimator (5) sometimes gets numerically unstable owing to the presence of small eigenvalues in the Hessian matrix.

For the quantitative comparison, we repeated the experiment by randomly changing the instance subsampling 100 times. Table 1 lists the average Kendall's tau and Jaccard index. The results again show that the proposed method can accurately estimate the true linear influences.

## 7.2 Evaluation on Data Cleansing

We now show that the proposed method is effective for data cleansing. Specifically, on MNIST [LeCun *et al.*, 1998] and CIFAR10 [Krizhevsky and Hinton, 2009], we demonstrate that we can effectively improve the models by removing influential instances suggested by the proposed method. The detailed setup and full results can be found in Appendix C.2 and C.4.

**Setup** We used MNIST and CIFAR10. From the original training set, we held out randomly selected 10,000 instances for the validation set and used the remaining instances as the training set. As models,

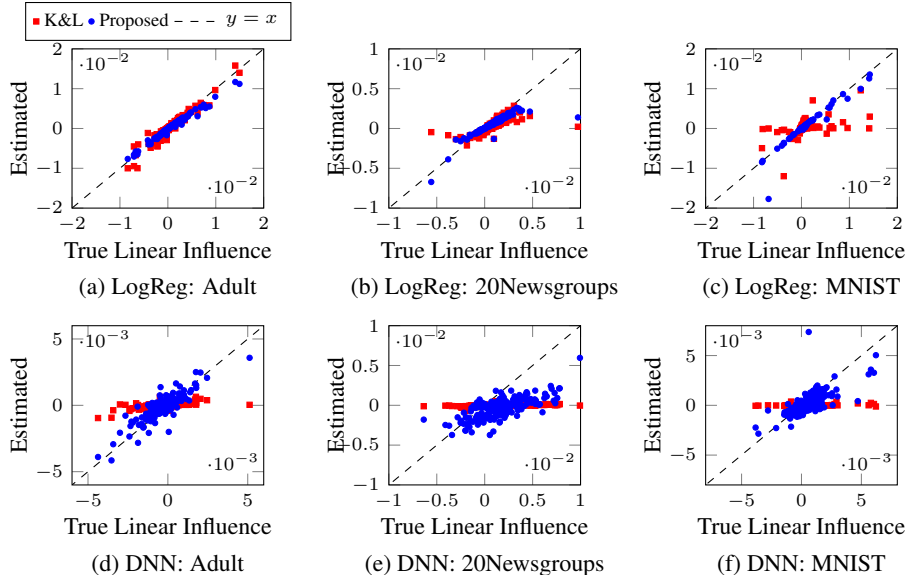

Figure 1: Estimated linear influences for linear logistic regression (LogReg) and deep neural networks (DNN) for all the 200 training instances. K&L denotes the method of Koh and Liang [2017].

Table 1: Average Kendall's tau and Jaccard index ($\pm$ std.).

| | Kendall's tau | | | | Jaccard index | | | |
|---|---|---|---|---|---|---|---|---|
| | LogReg | | DNN | | LogReg | | DNN | |
| | Proposed | K&L | Proposed | K&L | Proposed | K&L | Proposed | K&L |
| Adult | .93 (.02) | .85 (.07) | .75 (.10) | .54 (.12) | .80 (.10) | .60 (.17) | .59 (.16) | .32 (.11) |
| 20News | .94 (.05) | .82 (.15) | .45 (.12) | .37 (.12) | .79 (.15) | .52 (.19) | .25 (.08) | .11 (.07) |
| MNIST | .95 (.02) | .70 (.15) | .45 (.12) | .27 (.16) | .83 (.10) | .41 (.16) | .37 (.15) | .27 (.12) |

we used convolutional neural networks. In SGD, we set the epoch $K = 20$, batch size $|S_t| = 64$, and learning rate $\eta_t = 0.05$.

As baselines for data cleansing, in addition to K&L, we adopted two outlier detection methods, Autoencoder [Aggarwal, 2016] and Isolation Forest [Liu *et al.*, 2008]. We also adopted random data removal as the baseline. For the proposed method, we introduced an approximate version in this experiment. In Algorithm 2, the proposed method retraces all steps of the SGD. In the approximate version, we retrace only one epoch, which requires less computation than the original algorithm. Moreover, it is also storage friendly because we need to store intermediate information only in the last epoch of SGD.

We proceeded the experiment as follows. First, we trained the model with SGD using the training set. We then computed the influence of each training instance using the proposed method as well as other baseline methods. Here, we used the same query vector $u$ as in the previous experiment. Finally, we removed the top-$m$ influential instances from the training set and retrained the model. For model retraining, we ran normal SGD for 19 epochs and switched to counterfactual SGD in the last epoch.[8] If the misclassification rate of the retrained model decreases, we can conclude that the training set was effectively cleansed.

**Results**  We repeated the experiment by randomly changing the split between the training and validation set 30 times. Figure 2 shows the misclassification rates on the test set after data cleansing with each method.[9] It is evident from the figures that the misclassification rates decreased after data cleansing with the proposed method and its approximate version. We compared the misclassification rates before and after the data cleansing using t-test with the significance level set to 0.05. We

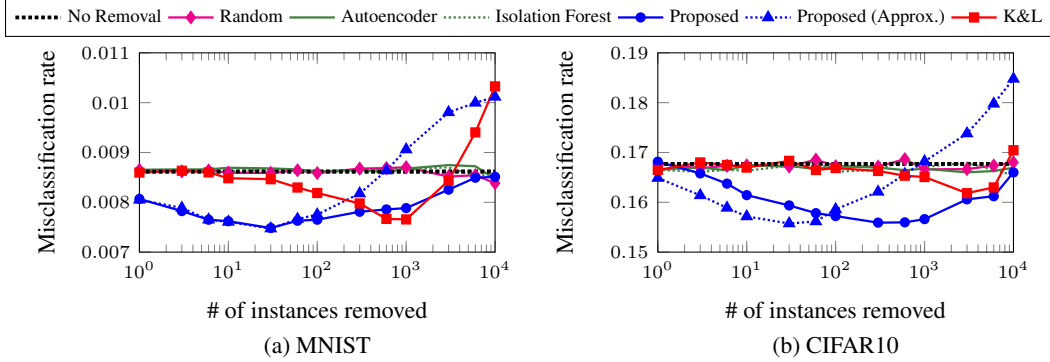

Figure 2: Average misclassification rates on the test set after data cleansing. The errorbars are omitted for better visibility. See Appendix C.4 for the full results.

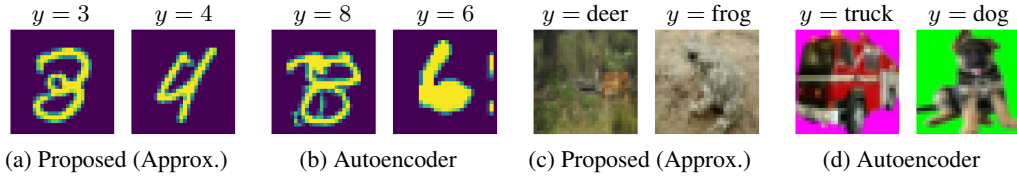

Figure 3: Examples of found influential instances and their labels in (a)(b) MNIST and (c)(d) CIFAR10.

observed that none of the baseline methods except K&L attained statistically significant improvements. By contrast, the proposed method and its approximate version attained statistically significant improvements. For both datasets, the proposed method and its approximate version were found to be statistically significant for the number of removed instances between 10 and 1000, and 10 and 100, respectively.[10] Moreover, both methods outperformed K&L. The results confirm that the proposed method can effectively suggest influential instances for data cleansing. We also note that the proposed method and its approximate version performed comparably well. This observation suggests that, in practice, we only need to retrace only one epoch for inferring the influential instances, which requires less computation and storing intermediate information only in the last epoch of SGD.

Figure 3 shows examples of found influential instances. An interesting observation is that Autoencoder tended to find images with noisy or vivid backgrounds. Visually, it seems reasonable to select them as outliers. However, as we have seen in Figure 2, removing these outliers did not help to improve the models. In contrast, the proposed method found images with confusing shapes or backgrounds. Although they are not strongly visually appealing as the outliers, Figure 2 confirms that these instances significantly affect the models. These observations indicate that the proposed method could find the influential instances, which can be missed even by users with domain knowledge.

# 8 Conclusion

We considered supporting non-experts to build accurate machine learning models through data cleansing by suggesting influential instances. Specifically, we aimed at establishing an algorithm that can infer the influential instances even for non-convex loss functions such as deep neural networks. Our idea is to use the fact that modern machine learning models are trained using SGD. We introduced a refined notion of influence for the models trained with SGD, which was named SGD-influence. We then proposed an algorithm that can accurately approximate the SGD-influence without running extra SGD. We also proved that the proposed method can provide valid estimates even for non-convex loss functions. The experimental results have shown that the proposed method can accurately infer influential instances. Moreover, on MNIST and CIFAR10, we demonstrated that the models can be effectively improved by removing the influential instances suggested by the proposed method.

**Acknowledgments**

Satoshi Hara is supported by JSPS KAKENHI Grant Number JP18K18106. Atsushi Nitanda is supported by JSPS KAKENHI Grant Number JP19K20337.

## Footnotes

[4]For Momentum-SGD, we can avoid storing the parameter $\theta^{[t]}$ [Maclaurin *et al.*, 2015].

[5]```grads = [tf.gradients(loss[i], theta) for i in St];        Hu = tf.reduce_mean(

[6]The codes are available at https://github.com/sato9hara/sgd-influence

[7]http://qwone.com/~jason/20Newsgroups/

[8]We observed that this works well. For the results with full counterfactual SGD, see Apendix C.4.

[9]See Appendix C.4 for the full results.

[10]See Appendix C.3 for a possible way to determine the number of removal in practice.

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
