[Supplementary Material]

## A    Relation to Koh and Liang [2017]

### A.1    Brief Review

As we mentioned in Section 6, Koh and Liang [2017] proposed to estimate the influence by (5), which is

$$\hat{\theta}_{-j} - \hat{\theta} \approx \frac{1}{N}\hat{H}^{-1}\nabla_\theta \ell(z_j; \hat{\theta}),$$

where $\hat{H} = \frac{1}{N}\sum_{z \in D} \nabla^2 \ell(z; \hat{\theta})$ is the Hessian of the problem (1) for the optimal model $\hat{\theta}$.

Note that, $\hat{H}^{-1}\nabla_\theta \ell(z_j; \hat{\theta})$ is equivalent to the solution to the following optimization problem:

$$\underset{\beta \in \mathbb{R}^p}{\mathrm{argmin}} \frac{1}{2}\langle \beta, \hat{H}\beta\rangle - \langle \nabla_\theta \ell(z_j; \hat{\theta}), \beta\rangle. \tag{6}$$

Koh and Liang [2017] proposed computing $\hat{H}^{-1}\nabla_\theta \ell(z_j; \hat{\theta})$ by solving this optimization problem using conjugate gradient descent or its improved version. In the optimization, they also proposed to use the mini-batch approximation of the Hessian matrix: they proposed to use $\hat{H}_S = \frac{1}{|S|}\sum_{z \in S} \nabla^2 \ell(z; \hat{\theta})$ on the mini-batch $S \subset D$ instead of the Hessian matrix $\hat{H}$ computed on the all training instances $D$.

### A.2    Relation to the Proposed Method

Here, we show the relationship between the proposed method and the method of Koh and Liang [2017]. Suppose that we solve the problem (6) using SGD. In the $t$-th step of SGD, we update $\beta$ by

$$\beta^{[t+1]} = \beta^{[t]} - \gamma_t(\hat{H}_{S_t}\beta^{[t]} - \nabla_\theta \ell(z_j; \hat{\theta})) = (I - \gamma_t \hat{H}_{S_t})\beta^{[t]} + \gamma_t \nabla_\theta \ell(z_j; \hat{\theta}),$$

where $S_t$ is the mini-batch and $\gamma_t > 0$ is a learning rate. Suppose that we initialized $\beta^{[1]} = \nabla_\theta \ell(z_j; \hat{\theta})$ and $\gamma := \max_t \gamma_t$. Then, the SGD for the problem (6) can be expressed as

$$\beta^{[2]} = (I - \gamma_1 \hat{H}_{S_1})\nabla_\theta \ell(z_j; \hat{\theta}) + \gamma_1 \nabla_\theta \ell(z_j; \hat{\theta}),$$
$$\beta^{[3]} = (I - \gamma_2 \hat{H}_{S_2})\beta^{[2]} + \gamma_2 \nabla_\theta \ell(z_j; \hat{\theta}) = (I - \gamma_2 \hat{H}_{S_2})(I - \gamma_1 \hat{H}_{S_1})\nabla_\theta \ell(z_j; \hat{\theta}) + O(\gamma),$$
$$\vdots$$
$$\beta^{[T]} = (I - \gamma_{T-1}\hat{H}_{S_{T-1}})(I - \gamma_{T-2}\hat{H}_{S_{T-2}})\dots(I - \gamma_1 \hat{H}_{S_1})\nabla_\theta \ell(z_j; \hat{\theta}) + O(\gamma).$$

Here, let $\hat{Z}_t := I - \gamma_t \hat{H}_{S_t}$, and we obtain

$$\hat{\theta}_{-j} - \hat{\theta} \approx \frac{1}{N}\hat{H}^{-1}\nabla_\theta \ell(z_j; \hat{\theta}) \approx \beta^{[T]} = \frac{1}{N}\hat{Z}_{T-1}\hat{Z}_{T-2}\dots\hat{Z}_1 \nabla_\theta \ell(z_j; \hat{\theta}) + O\left(\frac{\gamma}{N}\right).$$

When $\gamma$ is small and the last term is ignorable, this result resembles to the proposed estimator $\Delta\theta_{-j}$ in Section 4. Instead of $\hat{Z}_t := I - \gamma_t \hat{H}_{S_t}$ computed at the optimal model $\hat{\theta}$, the proposed estimator uses $Z_t = I - \eta_t H^{[t]}$ based on the model $\theta^{[t]}$ in the $t$-th SGD step in the training.

## B    Proof of Theorems

Before proving Theorems 5 and 6, we first prove the next lemma.

**Lemma 7.** Assume that $\ell(z; \theta)$ is twice differentiable with respect to the parameter $\theta$, and assume that there exist $\lambda, \Lambda > 0$ such that $\lambda I \prec \nabla_\theta^2 \ell(z; \theta) \prec \Lambda I$ for all $z, \theta$. If $\eta_s \leq 1/\Lambda$, then we get

$$h_j(\Lambda) \leq \|\Delta\theta_{-j}\| \leq h_j(\lambda), \tag{7}$$
$$h_j(\Lambda) \leq \|\theta_{-j}^{[T]} - \theta^{[T]}\| \leq h_j(\lambda), \tag{8}$$

where $h_j(a) := \frac{\eta_{\pi(j)}}{|S_{\pi(j)}|}\prod_{s=\pi(j)+1}^{T-1}(1 - \eta_s a)\|g(z_j; \theta^{[\pi(j)]})\|$.

*Proof.* Since $(1 - \eta_s\Lambda)I \prec Z_s \prec (1 - \eta_s\lambda)I$ we immediately obtain (7) from the definition (2) of $\Delta\theta_{-j}$.

We next show the inequality (8). There exists $r \in [0, 1]$ such that for $\theta_*^{[s]} := r\theta_{-j}^{[s]} + (1 - r)\theta^{[s]}$,

$$\frac{1}{|S_s|}\sum_{i \in S_s}\left(\nabla_\theta\ell(z_i; \theta_{-j}^{[s]}) - \nabla_\theta\ell(z_i; \theta^{[s]})\right) = H_*^{[s]}(\theta_{-j}^{[s]} - \theta^{[s]}),$$

where $H_*^{[s]} := \frac{1}{|S_s|}\sum_{i \in S_s}\nabla_\theta^2\ell(z_i; \theta_*^{[s]})$. Therefore, by setting $Z_s^* := (I - \eta_s H_*^{[s]})$, we can show the inequality (8) in a similar way to the proof of (7). □

## B.1 Proof of Theorem 5

*Proof.* From Lemma 7,

$$\|(\theta_{-j}^{[T]} - \theta^{[T]}) - \Delta\theta_{-j}\|^2 = \|\theta_{-j}^{[T]} - \theta^{[T]}\|^2 + \|\Delta\theta_{-j}\|^2 - 2\langle\theta_{-j}^{[T]} - \theta^{[T]}, \Delta\theta_{-j}\rangle$$
$$\leq h_j(\lambda)^2 + h_j(\lambda)^2 + 2h_j(\Lambda)^2 = 2(h_j(\lambda)^2 + h_j(\Lambda)^2).$$

□

## B.2 Proof of Theorem 6

*Proof.*

$$\theta_{-j}^{[s+1]} - \theta^{[s+1]} = Z_s(\theta_{-j}^{[s]} - \theta^{[s]}) + \eta(H^{[s]} - H_*^{[s]})(\theta_{-j}^{[s]} - \theta^{[s]}),$$

where $H_*^{[s]}$ is the same as that in the proof of Lemma 7. We set $D_s := \eta(H^{[s]} - H_*^{[s]})(\theta_{-j}^{[s]} - \theta^{[s]})$. Applying this equalities recursively over $s \in \{\pi(j), \ldots, T - 1\}$, we get

$$\theta_{-j}^{[T]} - \theta^{[T]} = \Delta\theta_{-j} + \sum_{s=\pi(j)}^{T-1}\prod_{k=s+1}^{T-1} Z_k D_s.$$

Hence, a remaining problem is to bound the norm of the second term in the right hand side of this equality, which corresponds to a gap we want to evaluate. Since $\|Z_k\| \leq 1 + \eta\Lambda$, $\|\theta_{-j}^{[s]} - \theta^{[s]}\| \leq 2\eta GT$ and $\|H^{[s]} - H_*^{[s]}\| \leq L\|\theta_{-j}^{[s]} - \theta^{[s]}\|$,

$$\left\|\sum_{s=\pi(j)}^{T-1}\prod_{k=s+1}^{T-1} Z_k D_s\right\| \leq \sum_{s=1}^{T-1}\prod_{k=s+1}^{T-1}\|Z_k\|\|D_s\| \leq \sum_{s=1}^{T-1}(1 + \eta\Lambda)^{T-s-1}\eta L\|\theta_{-j}^{[s]} - \theta^{[s]}\|^2$$

$$= 4\frac{(1 + \eta\Lambda)^{T-1} - 1}{(1 + \eta\Lambda) - 1}\eta^3 T^2 G^2 L \leq 4\frac{\left(1 + O(\gamma\Lambda/\sqrt{T})\right)^T}{\Lambda}\gamma^2 T G^2 L.$$

□

# C Details and Results of Experiments

## C.1 Setups in Section 7.1

**Datasets** We used three datasets: Adult [Dua and Karra Taniskidou, 2017], 20Newsgroups[6], and MNIST [LeCun *et al.*, 1998]. These are common benchmarks in tabular data analysis, natural language processing, and image recognition, respectively. We adopted these three datasets to demonstrate the validity of the proposed algorithm across different data domains.

We prepossessed each dataset as follows. In Adult, we transformed categorical features to numerical attributes [7]. In 20Newsgroups, we selected the two document categories `ibm.pc.hardware` and `mac.hardware`. As a preprocessing, we transformed the documents into numerical vectors using tf-idf, while removing frequent and scarce words. In MNIST, we selected the images from the two categories one and seven, so that the problem to be binary classification.

**Models** To see the validity of the proposed method beyond convexity, we adopted two models, which are linear logistic regression and deep neural networks. For linear logistic regression, we adopted the $\ell_2$-regularized loss $\ell(z;\theta) = \log(\exp(-y\langle\theta,x\rangle) + 1) + \frac{\alpha}{2}\|\theta\|^2$ where $y \in \{-1,1\}$. In the experiments, we determined the regularization parameter $\alpha$ using cross validation. For deep neural networks, we used a network with two fully connected layers each of which has eight

Table 2: Parameters used in SGD: $K$ denotes the number of epochs. $|S_t|$ denotes the batch size.

|  | LogReg | | | DNN | | |
|---|---|---|---|---|---|---|
|  | $K$ | $|S_t|$ | $\eta_t$ | $K$ | $|S_t|$ | $\eta_t$ |
| Adult | 20 | 5 | $\frac{0.1}{\sqrt{t}}$ | 10 | 20 | 0.1 |
| 20News | 10 | 5 | $\frac{0.01}{\sqrt{t}}$ | 10 | 20 | 0.1 |
| MNIST | 5 | 5 | $\frac{0.1}{\sqrt{t}}$ | 10 | 20 | 0.1 |

units with ReLU as an activation function. We used the sigmoid function at the output layer, and adopted the cross entropy as the loss function. To run SGD, we used the parameters shown in Table 2. We note that the loss function for the linear logistic regression is convex, while that for the deep neural networks is non-convex.

**Target Linear Influence** In the experiments, we randomly subsampled 200 instances for the training set $D$ and the validation set $D'$. We then estimated the linear influence for the validation loss using Algorithm 2. Here, we set the query vector $u$ as $u = \frac{1}{|D'|}\sum_{z'\in D'}\nabla_\theta\ell(z';\theta^{[T]})$. Estimation of the linear influence thus amounts to estimating the change in the validation loss

$$\langle u, \theta_{-j}^{[T]} - \theta^{[T]}\rangle \approx \frac{1}{|D'|}\sum_{z'\in D'}\left(\ell(z';\theta_{-j}^{[T]}) - \ell(z';\theta^{[T]})\right).$$

We note that the instances with large negative linear influences are deemed to be negatively affecting the resulting models. Removing such instances can improve the validation loss, and thus the users can prioritize the inspection of such instances.

**Baseline Method** We adopted the method of Koh and Liang [2017] as the baseline, which we abbreviated as K&L. In K&L, we estimate the influence by (5), which is

$$\hat{\theta}_{-j} - \hat{\theta} \approx \frac{1}{N}\hat{H}^{-1}\nabla_\theta\ell(z_j;\hat{\theta}),$$

where $\hat{H} = \frac{1}{N}\sum_{z\in D}\nabla^2\ell(z;\hat{\theta})$ is the Hessian of the problem (1) for the optimal model $\hat{\theta}$. For a query vector $u \in \mathbb{R}^p$, the linear influence can be estimated as

$$\langle\hat{\theta}_{-j} - \hat{\theta}, u\rangle \approx \frac{1}{N}\langle\hat{H}^{-1}\nabla_\theta\ell(z_j;\hat{\theta}), u\rangle = \frac{1}{N}\langle\nabla_\theta\ell(z_j;\hat{\theta}), \hat{H}^{-1}u\rangle.$$

Here, the last equality follows from the symmetricity of the Hessian matrix. Thus, for estimating the linear influence for all the training instances, we first compute $\hat{H}^{-1}u$, and then take an inner product with $\nabla_\theta\ell(z_j;\hat{\theta})$ for each training instance $z_j \in D$.

Note that, $\hat{H}^{-1}u$ is equivalent to the solution to the following optimization problem:

$$\underset{\beta\in\mathbb{R}^p}{\operatorname{argmin}} \frac{1}{2}\langle\beta, \hat{H}\beta\rangle - \langle u, \beta\rangle. \tag{9}$$

Koh and Liang [2017] proposed computing $\hat{H}^{-1}u$ by solving this optimization problem using conjugate gradient descent or its improved version. In the optimization, they also proposed to use the mini-batch approximation of the Hessian matrix: they proposed to use $\hat{H}_S = \frac{1}{|S|}\sum_{z\in S}\nabla^2\ell(z;\hat{\theta})$ on the mini-batch $S \subset D$ instead of the Hessian matrix $\hat{H}$ computed on the all training instances $D$. In the experiment, we used the implementation available at https://github.com/kohpangwei/influence-release.

**Evaluation Metrics** In the experiments, we ran the counterfactual SGD for all $z_j \in D$, and computed the true linear influence. We then used this ground truth to evaluate the goodness of the estimated linear influences. For evaluation, we adopted the following two metrics. The first metric is Kendall's tau. Kendall's tau is a typical metric for measuring ordinal associations between two observations. Kendall's tau takes the value between plus and minus one, where the value one indicates that the orders of the two observations are identical.

The second metric is Jaccard index. For data cleansing, the users are interested in instances with large positive or negative influences. We measured how accurately those important instances could be identified using the estimated influences. To this end, we selected 10 instances with largest positive and negative true influences, and constructed a set of 20 important instances. We compared this true important instances with the estimated important instances using Jaccard index. Jaccard index measures the similarity of the two sets. Jaccard index takes the value between zero and one, where the value one indicates that the sets are identical.

## C.2 Setups in Section 7.2

**Datasets** We used MNIST [LeCun *et al.*, 1998] and CIFAR10 [Krizhevsky and Hinton, 2009]. The MNIST dataset contains 60,000 training instances, while the CIFAR10 dataset contains 50,000 training instances. Both datasets also contain 10,000 test instanes. From the original training instances, we held out randomly selected 10,000 instances for the validation set, and used the remaining instances as the training set. Thus, in the experiment, we used 50,000 instances in MNIST and 40,000 instances in CIFAR10 for training, and the held out 10,000 instances for validation.

**Models** We used convolutional neural networks (CNNs) in the experiment. The network structures can be found in Figure 4. In SGD, we set the epochs $K = 20$, batch size $|S_t| = 64$, and learning rate $\eta_t = 0.05$. In the training, we used a simple data augmentation. For MNIST, we applied horizontal and vertical shifts in $\pm 2$ pixels. For CIFAR10, we applied horizontal and vertical shifts in $\pm 4$ pixels and horizontal flipping.

**Target Linear Influence** We set the query vector $u$ as $u = \frac{1}{|D'|} \sum_{z' \in D'} \nabla_\theta \ell(z'; \theta^{[T]})$. Estimation of the linear influence thus amounts to estimating the change in the validation loss

$$\langle u, \theta_{-j}^{[T]} - \theta^{[T]} \rangle \approx \frac{1}{|D'|} \sum_{z' \in D'} \left( \ell(z'; \theta_{-j}^{[T]}) - \ell(z'; \theta^{[T]}) \right).$$

We note that the instances with large negative linear influences are deemed to be negatively affecting the resulting models. Removing such instances can improve the validation loss, and thus the users can prioritize the inspection of such instances.

**Baseline Methods** For K&L [Koh and Liang, 2017], to solve the problem (9), we ran momentum-SGD for two epochs, where we set the learning rate to be 0.005, the size of momentum to be 0.9, and the batch size to be 1000. As baselines for data cleansing, in addition to K&L [Koh and Liang, 2017], we also adopted two outlier detection methods, Autoencoder [Aggarwal, 2016] and Isolation Forest [Liu *et al.*, 2008]. In outlier detection, we treated the validation set as a healthy dataset. We then computed outlierness of each training instance using outlier detection methods, as follows.

- **Autoencoder**: We trained an autoencoder using the validation set. See Figure 5 for the structures of autoencoders used. We adopted the squared loss as the training objective function. For training, we used Adam with the learning rate set to 0.001 and the batch size set to 128. We used the same data augmentation as the training of CNNs. After the autoencoder is trained, we fed each training input $x$ into the autoencoder and obtained an reconstructed input $\hat{x}$. We measured the outlierness of the input $x$ by $a = \|x - \hat{x}\|^2$.

- **Isolation Forest**: We first fed each validation input $x'$ into the trained CNN, and obtained its latent representation $r'$ from the flatten layer in Figure 4. We trained an isolation forest using the latent representations of the validation set. In the experiment, we used the `fit` method of `sklearn.ensemble.IsolationForest` with default configurations. After the isolation forest is trained, we fed each training input $x$ into the isolation forest and obtained its outlierness score $a$ using the `score_samples` method.

We also adopted random data removal as the baseline.

**Proposed Method** For the proposed method, we introduced an approximate version in this experiment. In Algorithm 2, the proposed method retraces the entire SGD steps. In the approximate version, we retrace only one epoch, which requires less computation than the original algorithm. Moreover, it is also storage friendly because we need to store intermediate information only in the last epoch of SGD.

**Procedure** We proceeded the experiment as follows. First, we trained the model with SGD using the training set. We then computed the influence of each training instance using the proposed method as well as the other baseline methods. Finally, we removed the top-$m$ influential instances from the training set and retrained the model. For the model ratraining, we considered the two settings.

- Retrain All: In this setting, we ran counterfactual SGD for all the 20 epochs with influential instances omitted.
- Retrain Last: In this setting, we ran normal SGD for 19 epochs and switched to counterfactual SGD in the last epoch with influential instances omitted.

If the misclassification rate of the retrained model decreases, we can conclude that the training set was effectively cleansed.

(a) CNN for MNIST      (b) CNN for CIFAR10

Figure 4: Structures of convolutional neural networks (CNNs)

(a) Autoencoder for MNIST    (b) Autoencoder for CIFAR10

Figure 5: Structures of Autoencoders

## C.3 Full Results in Section 7.2

The full results for MNIST are shown in Figures 6 and 7. The full results for CIFAR10 are shown in Figures 8 and 9. In the figures, it is evident that the misclassification rates have decreased after data cleansing with the proposed method and its approximate version. We compared the misclassification rates before and after the data cleansing using t-test with the significance level set to 0.05. We observed that non of the baseline methods have attained statistically significant improvements. By contrast, the proposed method and its approximate version attained statistically significant improvements: in MNIST, their improvements were statistically significant for the number of removed instances between 10 and 100, and in CIFAR10, their improvements were statistically significant for the number of removed instances between 100 and 1000. Figure 10 also confirms the effectiveness of the data cleansing with the proposed method. Out of 30 experiments, the misclassification rates decreased with the proposed method for 25 cases in MNIST, and for 26 cases in CIFAR10. These results confirm that the proposed method can effectively suggest influential instances for data cleansing. We also note that the proposed method and its approximation version performed comparably well. This observation suggests that, in practice, we only need to trace back only one epoch for inferring influential instances, which requires less computation and storing intermediate information only in the last epoch of SGD.

Figures 11 and 12 show the examples of found influential instances. An interesting observation is that Autoencoder tended to find images with noisy or vivid backgrounds. Visually, it seems reasonable to select them as outliers. However, as we have seen in Figure 2, removing these outliers didoes not help improving the models. On the other hand, the proposed method found images with confusing shapes or backgrounds. Although they are not strongly visually appealing as outliers, Figure 2 confirms that these instances have high impacts to the models. These observations indicate that the proposed method could find influential instances, which can be missed even by users with domain knowledge.

Figure 6: MNIST: Average misclassification rates on the test set after data cleansing over 30 experiments.

Figure 7: Exhaustive results on MNIST: [Thick lines] Average misclassification rates on the test set after data cleansing over 30 experiments. [Shaded Regions] Average ± standard deviation.

Figure 8: CIFAR10: Average misclassification rates on the test set after data cleansing over 30 experiments.

(a) Random

(b) K&L

(c) Autoencoder

(d) Isolation Forest

(e) Proposed

(f) Proposed (Approx.)

Figure 9: Exhaustive results on CIFAR10: [Thick lines] Average misclassification rates on the test set after data cleansing over 30 experiments. [Shaded Regions] Average ± standard deviation.

(a) MNIST

(b) CIFAR10

Figure 10: Comparison of the misclassification rates before and after the data cleansing with the proposed method. We set the number of removed instances to be 100 for MNIST and 10000 for CIFAR10.

(a) Autoencoder

(b) Isolation Forest

(c) Proposed

(d) Proposed (Approx.)

(e) K&L [Koh and Liang, 2017]

Figure 11: Examples of found top-20 influential instances in MNIST

(a) Autoencoder

(b) Isolation Forest

(c) Proposed

(d) Proposed (Approx.)

(e) K&L [Koh and Liang, 2017]

Figure 12: Examples of found top-20 influential instances in CIFAR10

## Footnotes

[6]http://qwone.com/~jason/20Newsgroups/

[7]We used the implementation available at https://www.kaggle.com/kost13/us-income-logistic-regression/notebook