[Reviews · NeurIPS 2019]

Reviewer 1



[Update] I read the authors' reply and the other reviews. Reviewer 2 raises some important points and I agree with some of the concerns about the clarity of the paper and the contrast to existing other work. The authors' reply contains some additional experiments, which is great. And I appreciate the clarification of my misunderstanding of the storage of parameters, not gradients. [Update] I enjoyed reading the paper, the authors do a great job at motivating their approach and describing the derivation. Data cleansing is an important step in machine learning systems and principled approaches for automating this step is a valuable contribution. The setting considered by the authors covers a substantial fraction of ML algorithms used today. Hence the proposed method has the potential to improve data cleansing for most models being used in practice. Overall the manuscript is written in a clear and concise manner. Also the experimental section is solid for the most part. I had a few minor comments and some concerns about the experimental settings. The supplementary material is very helpful, in particular it is impressive that the authors provide two implementations, in tensorflow and pytorch. Originality The method appears to be novel and the approach is well embedded into the related work. What wasn’t really clear to me is how the proposed approach is an improvement over just rerunning the SGD. In the proposed algorithm 1 and 2, it seems the approach requires to store the gradients of every step in the SGD optimisation? This is a lot of data that needs to be stored, more than the actual data set, right (or are the gradients only stored for one epoch)? If that’s the case then running the naive version of leaving one data point gradient out amounts to a mere sum over the gradients, which is a trivial and efficient reduce operation. Quality I found the quality of the paper to be higher than many papers I’ve reviewed for this venue. The paper is well written and motivated, the supplementary material is extensive and the experimental section demonstrates clearly the advantage of the method compared to other baselines for data cleansing. There was one thing that stood out in the comparisons. In Section 7.1 the authors compare agains a method that requires convex losses, but the models used in the comparisons have non-convex losses. In order to render the baseline method usable for Hessians with negative eigenvalues, the authors simply add a positive constant of 0.1 to the Hessian. Even though the authors of the baseline method suggest this, they probably suggested this as a remedy for numerical instability under the assumption that the Hessian should actually not have negative eigenvalues - and not as a way to render their method usable for non-convex losses. So I’m wondering how fair a comparison this is. Overall, it seems that in these experiments the proposed method really only shines in the case of the MNIST data, in the other cases the difference is not too large. If one could exclude that this MNIST case was due to a wrongly calibrated offset for the Hessian, this would render the comparisons much stronger, no? Clarity Overall the paper is very clear and concise. There are just some minor things I feel that could be explained better. Line 96: The role and importance of the query vector is not entirely clear to me - is that needed for not having to compute the Hessian explicitly or is it just a common approach? Line 163: I’m not sure this is obvious, but to me it wasn’t clear, how the Hessian times u is equivalent to the derivative of the inner product of the query vector and the gradient of the loss. This appears to be one of the central contributions and could deserve some more attention in the manuscript? Significance The problem of data cleansing and its automation is highly relevant and this manuscript appears to be a valuable contribution in that space. The authors provide solid experimental validation of the proposed method in a data cleansing application.

Reviewer 2



=== UPDATE === After reading the author feedback and other reviews (and re-reading the submission and [K&L, ICML17]), I will make a few points of clarification regarding my understanding of the paper. - I do better understand how this is not anomaly detection (although anomalous points intuitively often incur high loss, so this would likely be correlated in many cases) but, as you state, Figure 2 shows that this is different in a practical way, the counterfactual estimation is clearly a conceptual difference, and Figures 11,12 are interesting. However, I would suggest making it clear that this is not an anomaly detection algorithm (at least in the traditional sense) in Section 7.2 discussion. - I now see that lines 233-234 literally means Section 4.2 of of [K&L]. However, to my point regarding a clear contrast, this paper should read as (1) [K&L] proposed a method for using influence functions that can be used for data cleaning, (2) They suggest adding a diagonal regularizer is effective for non-convex settings [maybe a sidebar about the methods that follow], (3) we operate directly on SGD, (4) we have an efficient estimator of this, (5) it works. Between translating notation and claims between this submission and [K&L], I didn’t get this when reading — as I found the precise setting confusing in general. Now that I look closer from this perspective, I think there is less novelty than the submission implies, but more than I originally thought (and that it weakly meets the NeurIPS bar in this regard). Accordingly, I am a bit more convinced regarding the method itself and that there is something possibly important in here — even if not fully developed. In this vein, my technical concerns remain that additional datasets (e.g., Figure 6 of [K&L] would help, Theorem 5 isn’t important as the convex case was already understood, Theorem 6 isn’t useful (i.e., more of a case of what can be proven than what we want to know), and Lines 157-177 are also closely related to section 3 of [K&L]. Also, I still think the paper is sort of mess in its current form wrt the narrative structure, although other reviewers seem confident this can be fixed before the camera-ready deadline. My overall assessment basically remains the same, while I am more confident that there is some interesting idea here, I lean toward them requiring further development before publication === The authors build on recent work regarding interpretability based on influence functions [Koh & Liang, ICML17] to identify instances that should be censored during classifier training in order to obtain a better performing model. Methodologically, they propose SGD-influence, a method that considers the effect on the model of censoring specific examples (i.e., a counterfactual consideration) and develop an efficient influence estimation to develop the data cleaning procedure. Experiments are conducted on Adult, 20Newsgroups, MNIST, and CIFAR10 — showing that the linear approximation is a sufficient estimation to the true counterfactual influence of specific examples and that removing influential examples can improve performance (different datasets for different tasks), contrasting with a baseline based on [Koh & Liang, ICML17]. Overall, cleaning up datasets and theoretical properties of noise-tolerant learning are long-studied areas. The use of influence functions is timely and ostensibly promising empirically — and I appreciate two families of experiments on datasets from different domains. However, I also have several concerns with this submission as itemized below. - First, the setting described in the abstract/introduction doesn’t exactly match what I believe is performed. Specifically, in line 5, “users only need to inspect the instances suggested by the algorithm” and in lines 32-33, “they only need to inspect the instances suggested by the algorithm”. These statements imply a human in the loop procedure (e.g., something like [Rebbapragada, et al., Active Label Correction, ICDM12] or [Siddiqui, et al., Feedback-Guided Anomaly Discovery via Online Optimization, KDD18]). However, it doesn’t seem this is actually the case as the top-m are removed without an audit procedure. Secondly, it there was a human in the loop, I am not entirely convinced that it wouldn’t require domain knowledge to accomplish this effectively (e.g., Figure 12 in the Appendix; I would have no idea which to remove). While this doesn’t have an impact on the technical results, this should be clarified. - In a similar vein, this work isn’t sufficiently contextualized wrt more traditional related work. From a conceptual perspective, it really isn’t clear if what is actually being accomplished is noise-tolerance (e.g., [Aslam & Decatur, On the sample complexity of noise-tolerant learning, 1996], [Natarajan, et al., Learning with Noisy Labels, NeurIPS13]), dataset (label) cleaning (e.g., [Brodley & Friedl, Identifying Mislabeled Training Data, JAIR99], or anomaly detection. The implication is that it is anomaly detection as this is what is done empirically, but this isn’t entirely clear to me. Secondly, work on active learning via maximum model change (e.g., [Settles, Craven & Ray, Multiple-Instance Active Learning, NeurIPS08], [Cai, et al., Active Learning for Support Vector Machines with Maximum Model Change, ECML14], [Cai, et al., Active Learning for Classification with Maximum Model Change, TOIS17]) is also relevant — especially if you are actually doing active learning. Finally, it seems the methodological comparison is to Section 2,3 of [Koh & Liang, ICML17] while section 4 discusses relevant extensions (that are mentioned in the Appendices) and [Zhang, et al., AAAI18], [Khanna, et al., AISTATS19] provide extensions that are not discussed in any detail. Without reading these papers, it isn’t clear precisely what the contribution is. Honestly, even after reading them, it really isn’t particularly clear (beyond high-level statements) — and this extends all the way through experiments, etc. - Also from a clarity perspective, the work pretty much requires reading the Appendices. Specifically, dataset sizes, m in ‘top-m’, sketches of proofs, etc. all require significant decoding to precisely understand. Beyond this, I would also recommend moving ‘influence estimation’ earlier in the paper (e.g., at the end of section 2) as it pretty much is the starting point for distinguishing this work as far as the narrative goes. I know that there are space constraints, but this paper is very difficult to parse without reading several times and reorganizing into a more linear narrative. Honestly, in reading the related works and this material, it seems that this work is largely fleshing out specific sections of [Koh & Liang, ICML17] — I might be underestimating the contribution, but it isn’t clear from the text. - As previously mentioned, while I think the number of datasets are sufficient and the results seem promising, it isn’t clear if the correct baselines are being used (e.g., equations from sections 4.2 and 4.3 of [Koh & Liang, ICML17]). Additionally, some of the details from the Appendix have to make their way into the main paper and there needs to be discussion of the empirical results beyond ‘it works better’ than a given baseline. Thus, while I think this is an important problem and I am fairly certain there are some nuggets in this submission to build on, this work is not sufficiently developed in its current form to be accepted at a top-tier conference. A clearer description of the setting, better contextualization of this work from a conceptual perspective, more precise contrast with the most directly related work, and more discussion of the empirical results would make this a stronger submission. Below, I will rephrase/elaborate a bit more exhaustively along the specifically requested dimensions. === Originality Even after reading the most directly related work (e.g., [Koh & Liang, ICML17], [Khanna, et al, AISTATS19], [Zhang, et al., AAAI18]) and the appendices, it isn’t precisely clear how much overlap there is between the direct non-convex extension here and how others have handled this. While I can see the difference from a math perspective, I am not certain what is the effective difference from a practical perspective and this is really the job of the authors to state. In the absence of these works, using ideas around expected gradient length, expected model change, etc. are well-studied in the active learning literature — so it isn’t a large conceptual leap. Thus, my assessment regarding originality is that it doesn’t meet the NeurIPS bar, but I might be wrong if I am not understanding the precise contribution due to the narrative of the paper. === Quality Once I went through the appendices, the contribution made a bit more sense as they included proofs and details of the empirical study. Overall, the paper should be more self-contained (even with space restrictions). I think the proofs can be omitted, but a few sentences regarding the contribution relative to other works and contrasting the results would be helpful. As I stated above, if the most related work was introduced earlier in the paper, the math is easier to comprehend (that I how I had to do it on my own piece of paper). Basically, the math and methods should be pre-digested for the reader. For the empirical results, I think there are a sufficient number of datasets, but the work would benefit from more comparisons to the most recent work (and at least discussion regarding other sections of [K&L, ICML17]). Additionally, a few more details have to make their way into the paper (e.g., the m in top-m) and it would help to present some of the statistics (e.g., data size), etc. in a table or other easily referenced section. === Clarity Summarizing points from other sections, the math should be presented such that the precise contribution can be easily seen and the empirical results should be restructured such that the conclusions are more self-evident (although there should also be more discussion). However, as stated previously, my most significant confusion regarding clarity is the mismatch between the motivation/introduction where non-experts are auditing suggested examples and the paper where I believe examples are automatically thrown away. Of course, this doesn’t affect the technical contribution (unless there is some auditing procedure that isn’t being discussed) — but the the paper could benefit from moving some of the appendix material, additional discussion regarding empirical results, additional contextualization wrt related work and instead this space is being spent misstating the setting. === Significance Automatic data cleaning is an important problem and Figure 2 looks interesting in this regard. Thus, I am convinced that there is a solid, usable contribution in here. However, without a clear contrast with other parts of [Koh & Liang, ICML17] and works that follow, I am unsure if this is a fair comparison. I think that the result, if made clearer and more convincing is likely sufficiently significant.

Reviewer 3



****Update**** Thanks to the authors for the clarifications. After reading the other reviews, I still stand by my score. I believe the method and the insights are potentially useful to the community. =========================================================== The paper deals with an important problem of estimating influence estimation. Tackling the problem has implications across modern day machine learning community. The proposed influence estimator leverages the recursive SGD formulation to approximate the proposed influence function. The proposed method is interesting and potentially impactful. The problem is nicely setup and writing is clear. The experimental evaluation is sufficient to show the promise of the proposed method The error bound of \delta \theta_{-j} for non-convex objectives obviously becomes loose as T increases. The final influence estimates for the DNN experiments are understandably bad, compared to logreg. However, it would be interesting to know how good/bad was the influence approximation as the training proceeded. Could you please comment on this?

Reviewer 4



originality: high, the proposed method for the SGD estimation is novel, while the second-order approximation is standard but reasonable. quality: high, the theoretic analysis and algorithm looks reasonable clarity: high, paper writing is clear significance: medium, the experiment is not as excited as the proposed method and analysis. Therefore, it may not show practical usage now, but this direction may bring out some work that can be improved, as it is an important question. -----------------------------Post-Rebuttal-------------------------------------- After the authors addressed my question in the amount of data to remove, I raised my score to 7. The main contribution of the paper is to provide an efficient algorithm to estimate the influence function better when using SGD updates, which is useful to the field since most measures (such as influence function) in Deep learning does not consider the optimization procedure. The framework can also provide decent theoretic analysis, and potentially can be improved with better high-order approximations. The experiments are not very practical useful yet (compared to other data cleansing methods), but the method proposed is still useful to the community.

[Author Response · NeurIPS 2019]

We thank the reviewers for their constructive feedback, especially for Rev#2 and #4 for
suggesting related studies. We will improve the readability of the paper, and enrich the
discussion with suggested related studies. Please find below the answer to the reviews
(apologize it is not exhaustive due to space limitation).

Figure 1: DNN: Adult

**Rev#1: mere sum over the gradients** The suggested estimator is expressed as $\Delta\theta_{-j} =$
$\sum_{k=1}^{K} \frac{\eta_{\pi_k(j)}}{|S_{\pi_k(j)}|} g(z_j; \theta^{[\pi_k(j)]})$. Figure 1 shows that the suggested estimator did not work well
for the adult dataset with DNN in the experiment in Sec7.1, which suggests that the use of
Hessian in the proposed estimator is essential. We will add this result in the future version.
Also, please note that we store the model parameter $\theta^{[t]}$ but not the gradients in each step of
SGD. The stored parameter is first loaded to the model, and then the gradient for each instance is computed.

**Rev#1: fair comparison?** [Koh & Liang] mentioned that their estimator (with the diagonal regularization) is effective
for DNNs. The comparison is therefore essential to show that our estimator is more suitable for DNNs.

**Rev#2: human-in-the-loop?** We are grateful for a suggestion for clarifying our contribution. Our method is automatic
and does not require user intervention. We will update the abstract/introduction to be aligned with the proposed method.
We mentioned the user intervention because that is a common way the most data scientists do for data cleansing.

**Rev#2: relation to noise tolerance, label cleansing, anomaly detection** We appreciate you for raising so many
related research topics. Anomaly detection looks for instances away from the data distribution, however, it is not
guaranteed that such instances are influential to the model performance. The proposed method directly looks for
instances that minimize the validation loss. The results in Sec7.2 confirm that this direct approach is more effective.
The noise tolerant learning and label cleansing will be interesting related topics. We will cite references and discuss
them as related studies. The difference from our study is that these studies assume that the label noise is an only issue.
However, as Figs 11 and 12 show, the model performance depends not only on label noises but atypical inputs also. For
example, in Fig11, we can find several atypical instances that even human cannot label them confidently. These atypical
instances should be removed from the training rather than fixing the labels because we cannot put correct labels to them.

**Rev#2: how others have handled non-convexity** To our knowledge, none of the raised studies can handle non-
conveixty. Capability of handling non-convexity is therefore our notable contribution. [Koh & Liang; Zhang, et al.]
assume convexity explicitly, while [Khanna, et al] does not (but it is implicitly assumed in Proposition 3). The active
learning studies [Settles, Craven & Ray], [Cai, et al.], [Cai, et al.] are evaluated only on convex problems such as SVM
and logistic regression. The theoretical analysis given by [Cai, et al.] is limited to these convex problems also.

**Rev#2: clear contrast with [Koh & Liang] and works that follow** We adopted [Koh & Liang] as the baseline and
excluded [Zhang, et al.] and [Khanna, et al] because (i) [Zhang, et al.] is devoted for label collection, which is different
from our goal (i.e. removing harmful instances); (ii) The method of [Khanna, et al] is computationally very expensive,
which requires computing the inner product $\langle\nabla_\theta \ell(z_i; \theta), \nabla_\theta \ell(z_j'; \theta)\rangle$ for all the training instances $z_i$ and for all the
validation instances $z_j'$ (the time complexity is at most $O(N^2)$). We will clarify this point in the future version.

**Rev#3: DNN experiments are understandably bad compared to logreg** For the non-convex case, as Theorem 6
shows, the estimation error can grow as the SGD steps $T$ gets large. In our preliminary experiment, we observed
this holds true in practice. The less accurate estimation compared to logreg in Fig1 and Tab1 is therefore a natural
consequence. Constructing a more accurate estimator would be an important future direction.

**Rev#4: overfitting problem** This is a consequence from the definition of SGD-influence. SGD-influence (as well
as ordinary influence) considers removing only one instance, and ignores the higher-order interactions between the
instances. Extending the method for multiple instances is an important future direction. We will clarify this point.

**Rev#4: how much data to be removed** A reasonable solution would be selecting
the number of removal that minimizes the error on an additional set. Note that this
additional set is used only for determining a single scalar, and we do not need many
instances for this purpose. In both MNIST and CIFAR10, we observed that preparing
only 500 instances would be sufficient. See Figure 2 for the case of MNIST (shade =
variation among random repetitions). We will add this result in the future version.

Figure 2: Removal on MNIST

**Rev#4: related works [1, 2, 3]** We appreciate you for raising related studies. Unfor-
tunately, the reference [3] was missing in the review, and we therefore checked only
[1] and [2]. First of all, we are happy to find out that several different approaches are studied. We will include them
as related works in the future version. The advantage of our study is in theories of the estimation error. We believe
establishing solid theoretical foundation is essential to move the entire field forward. We hope our study to be a first
step towards establishing further principled and sophisticated algorithms for automatic data cleansing in the future.

[Meta-Review · NeurIPS 2019]

The paper addresses the problem of estimating the influence of a data point, for models trained with SGD, for non-convex losses. It then describes a data cleaning method based on this estimate. Reviewers found the theory solid, and the proposed data-cleaning method promising.